# Impact Assessment Model for the Implementation of Cargo Bike Transshipment Points in Urban Districts

**Tom Assmann [1,*], Sebastian Lang [2], Florian Müller [3] and Michael Schenk [1]**

[1]  Institute of Logistics and Material Handling Systems, Otto von Guericke University, 39106 Magdeburg, Germany; michael.schenk@ovgu.de

[2]  Fraunhofer Institute for Factory Operations and Automation IFF, 39106 Magdeburg, Germany; sebastian.lang@iff.fraunhofer.de

[3]  Institute of Psychology, Otto von Guericke University Magdeburg, Universitätsplatz 2, 39106 Magdeburg, Germany; florianjy.mueller@posteo.de

*  Correspondence: tom.assmann@ovgu.de; Tel.: +49-391-67-52627

**Abstract:** Mitigating climate change and improving urban livability is prompting cities to improve sustainability of urban transportation and logistics. Cargo bikes, in combination with urban transshipment points, are gaining momentum as a green last mile alternative. Although a wide body of research proves their viability in dense urban areas, knowledge about planning urban transshipment points is very limited. This also entails the siting of such facilities and the assessment of effects on emissions. This study therefore presents a first quantitative scenario-based model that assesses the impacts on a district. It examines different strategies for siting urban transshipment points in a single district and its effect on traffic, the carbon footprint, and air quality to give strategic insights where to create candidate locations for such facilities. Our result contributes to knowledge of planning urban transshipment facilities and assessing the impact of different configurations. The findings demonstrated that the use of cargo bikes to make courier, express, and parcel (CEP) deliveries in urban districts could reduce greenhouse gas (GHG), particulate matter (PM10), and nitrogen oxides (NOx) emissions significantly. However, the choice of vehicles completing inbound and outbound processes and the strategies for siting urban transshipment points display widely differing and even conflicting potential to reduce emissions.

**Keywords:** urban logistics; cargo bike; urban transshipment point; city logistics; urban planning; urban freight

## 1. Introduction

As the greening of urban logistics is increasingly becoming a priority for cities all over the world in these days of global warming and urbanization, logistics providers and municipalities must quickly produce new concepts for the last mile, which has long been overlooked [1,2]. Urban and private planners are emphasizing the incorporation of cargo bikes in sustainable urban mobility plans [3,4]. Courier, express, and parcel (CEP) services are increasingly collaborating with municipalities to implement small urban transshipment points (UTPs), sometimes called satellites [5]. Well-functioning examples [6,7] demonstrate their cost effectiveness and potential to cut greenhouse gas (GHG) emissions.

Promoting the use of cargo bikes systems as the dominant logistics network for distinct urban districts requires the provision of land for UTPs. Interviewed industry experts indicate that the establishment of such facilities is primarily contingent on identifying suitable sites since land is limited and subject to competition for different potential uses [8]. While municipalities must provide land for essential infrastructure, they have virtually no guidelines for identifying suitable candidate sites

for UTPs. Different population densities and business populations dictate the freight requirements in a district [9] and thus affect logistics networks significantly. The problematic situation of siting UTPs is exacerbated by the lack of a one-size-fits-all solution for districts with different population densities and business populations. This means, for instance, that public or industry planners have no information on whether siting a UTP in a district's center or on its periphery, or siting one large UTP or multiple smaller ones is better in terms of traffic and emissions.

## 1.1. Literature Survey and Research Gap

Urban logistics research and solutions for its sustainability is a trendy research topic [10]. Although public authorities form a dominant perspective in urban logistics research [11], most work is focusing innovations, best-practices, and technical solutions on a small scale, often neglecting environmental and social aspects [10].

Cargo bikes are emission-free, environmentally friendly vehicles [3]. Furthermore, they constitute a viable means of transport in urban delivery systems [12] and can cut costs of delivery [6,13]. Their logistical benefit arises from the potential to reduce driving time over short distances in dense urban settings, especially with higher levels of congestion [14]. Thus, many field trials and operational cargo bike delivery networks are deployed by CEP providers. Although UTPs are an essential component, their strategic siting is mostly neglected and candidate sites are not available [15]. One strain of research at this point are planning handbooks. Stiehm et al. [16] provide profound insights into layouts and corresponding dimensions of internal areas for logistics operations, legislative aspects, and emphasize an active role of urban planning. The handbook of Assmann et al. [15] also deals with UTP siting, but concentrates on the planning process, acceptability, and cycling infrastructure. They favor cooperative UTPs (cUTPs), which are jointly used by several CEP-companies. Both publications clearly outline the need for a strategic planning of UTPs and the provision of corresponding space by cities.

Cargo bikes are generally seen as a means to cut emissions. Concerning GHG emissions, this is even the case when the deployment of cargo bikes worsens speed and increases delay times as Melo and Baptista [17] demonstrate. A microscopic traffic simulation indicates that improvements can be achieved until the threshold of cargo bikes substitutes approximately 10% of delivery vans. However, the spatial frame covers just one street rather than a whole district. Assmann et al. [15] demonstrate by means of microscopic traffic simulation that more than 20 cargo bikes an hour in small streets used by bikes and vehicles in equal rights decrease quality of traffic flow. This changes when the effect of parked cargo bikes for delivery on traffic flow is compared to parked vans on the road. Cargo bikes, thanks to smaller dimensions benefit traffic flow and hence emissions. Unfortunately, those findings do not draw a coherent picture of cargo bikes' impacts on traffic and emissions.

Research focusing the scale of a district reveals more coherent results. A straightforward approach to determine the UTP location is to minimize transport, e.g., by means of simulation [18]. Arvidsson et al. [19] examined a mobile-hub delivery network. They demonstrate fairly linear reduction potentials for GHG-emissions. Air pollution is not assessed, but decreasing space occupancy is highlighted. Important to note is the evaluation of space in spatial parameters and not distance related parameters. Arnold et al. [20] investigated the interplay of delivery costs and external costs for different delivery networks. They propose a mixture of self-pick up and bike delivery to find the best balance between both factors. Real-world field trial evaluations of cargo bike delivery networks [7,21] and case specific ex ante calculation [22] reveal a GHG-reduction potential too. However, no general conclusions can be drawn from all these findings above. The footprint methodology is city-independent. Shahmohammadi et al. [23] examine three different delivery supply chains in which cargo bikes show a strong potential for decreasing GHG-emissions. Rudolph et al. [24] take a case-independent approach of looking at delivery networks. They demonstrate that UTPs with cargo bikes can decrease overall kilometrage and emissions in districts better than conventional delivery networks by applying an analytical spreadsheet model. However, their reduction in kilometrage does not meet the empirical

findings of mentioned field trials. As they do not vary the population density, they fail to deliver any strategic conclusions for planners.

Taking more general approaches of planning UTPs into account, the assistance for urban planners to strategically plan sites for UTPs remains scarce. Location routing problems (LRPs) rely on a set of candidate sites [25,26]. Although they would allow assessments of emissions [27,28] such approaches do not help planners solve the existing problem of where to create candidates. The interplay between fleet size, UTP location, and emissions is studied by Koç et al. [27], showing a strong effect of urban form and its related travel speed on GHG-emissions. However, the underlying urban model is not a good fit for urban forms of European cities and their districts. Multi-criteria/multi-attribute decision models can be used to determine UPT locations under uncertainty and can take quantitative and qualitative attributes into account [29]. Rao et al. [30] demonstrate this methodology aiming for sustainability but still need given candidate locations. Frameworks to evaluate urban logistics schemes are introduced as an ex ante simulation by Heeswijk et al. [31], not narrowing down to urban districts, as transferability framework [32], and for implemented schemes [33,34]. None of them provide a sound emission calculation.

The literature study clearly outlines a missing methodology assisting urban planners to strategically define candidate locations for UTPs. Current approaches are based on given candidate locations, but what is needed is guidance in which area such candidates should be made available, and also for large substitution levels of vans for cargo bikes. To assist this process the planner should be able to assess potential effects on traffic and environment of those candidates, not just in qualitative but quantitative terms. A universal methodology for siting UTPs in urban districts is clearly lacking in the literature on cargo bike systems. Until now, the focus has been on examining certain good practices that demonstrate the benefits of cargo bikes.

*1.2. Scientific Contribution, Methodology, and Structure of the Paper*

An impact assessment of the aforementioned situation on how and where to site a UTP is therefore needed to assist government and industry. The objective of this study is to assess ex ante the impacts on traffic, carbon footprint, and air pollution when a cargo bike delivery network with different strategies for siting UTPs is implemented in one entire district. Thus, our work will contribute to research on assessing and planning urban cargo bike delivery networks. We aim to provide a mathematical, scenario-based model. It shall deliver generic, strategic insights for researchers, policymakers (including planners), and logistics service providers (LSP)s.

Models or modelling methodologies are required to evaluate urban logistic schemes [35]. In the following, we will present a model to represent the cargo bike delivery network. When developing it, the procedure model for a simulation according to the Association for Simulation in German speaking countries (ASIM) [36,37] is applied. Our modification is to remain static in relation to time, as explained later. The methodology knows eight phases in the course of work. On the one hand, five of them directly build the model, namely: Objective and task definition, conceptual model, formal model, implemented model, and lastly, experiments and results. On the other hand, data gathering, data processing, and validation complete it. We structure the paper accordingly.

The gap in scientific research and the resulting objective and task are defined in Section 1. The conceptualization provides an understanding of the system and the reasoning for model construction decisions. This is described in Section 2. Then, Section 3 introduces the mathematical (formal) model and briefly highlights its implementation. The gathered and processed data is provided subsequently. We present our results in Section 5. A discussion and conclusion end the paper.

## 2. Conceptual Formulation to Model Urban Districts and Urban Transshipment Point (UTP) Siting Strategies

The model presented in this paper assumes that a municipality desires to shift the majority of last mile deliveries in an inner-city district (delivery area) from vans to cargo bikes. CEP service has been

selected as the scope of analysis since it is ideally suited for delivery by cargo bikes. As is the case in Germany, the market consists of five companies with different market shares. The model assesses total kilometrage, GHG, nitrogen oxides (NOx), and particulate matter (PM10) emissions in the delivery area, and the inbound and outbound processes of a cargo bike system with different UTP strategies. Conventional deliveries by vans constitute the reference case (Figure 1). The inbound and outbound processes are the deliveries to the UTPs from the CEP services' conventional hubs, predominantly located at the periphery of the urban area [6,38].

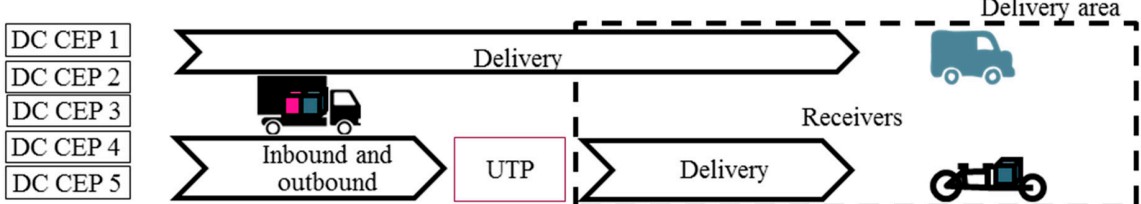

**Figure 1.** Courier, express, and parcel (CEP) delivery system with conventional vans and with urban transshipment points (UTPs) and cargo bikes.

Urban planning is a discipline characterized by long planning horizons, but still needing to make decisions on future urban form. Therefore, the model shall represent average workdays in the year 2025. Representing an average day allows aggregation to monthly and annual values. Since it must be simple enough so that experts in domains other than logistics modeling understand its methodology and reasoning, the model is based on a generalized urban form (Section 2.1), aggregated demand, and delivery data (Section 4).

### 2.1. Urban District

Cities are divided into districts, which are divided into quarters [39]. A quarter is an urban planning unit denoting a European inner-city area of approximately 1 km$^2$ with 10,000 to 35,000 residents per km$^2$ and a specific but not dominant business population (business, retail, etc.) [39]. The model uses two business population scenarios—one in a residential quarter and one in a mixed-use quarter as specified by the German Zoning Ordinance [40]—to assess the impact of population density in the designated range.

The model builds on two contiguous quarters that form one district. A Manhattan distance is used to simplify their modeling as a square. Industry experts maintain that the maximum distance between a UTP and customers should be no more than 1 km [8]. Since the model assumes that the population is distributed homogeneously, freight demand is evenly distributed geographically.

### 2.2. Strategies for Implementing Urban Transshipment Points

This paper defines UTPs as facilities where one or several logistics providers transfer their deliveries from conventional vehicles to cargo bikes. Freight is neither bundled nor stored there (short term) as it is in consolidation facilities. There are singular UTPs (sUTPs) where one logistics provider uses a single location and there are cooperative UTPs (cUTPs) where several logistics providers use one location together; the freight flows in the latter are strictly separated. Whereas cUTPs are meant to be stationary, sUTPs can be stationary, semi-stationary, or mobile [6,41].

Siting a UTP entails dealing with the dilemma of locating it as centrally as possible to minimize delivery distance while keeping it outside the district to minimize the impact of inbound and outbound traffic. The following UTP strategies (STR) are prone to further assessments (Figure 2):

1.  U-C: The cUTP is stationary and located at the main approach to the district. There is one delivery zone of 2 km$^2$.

2.  U-Q: Each of two stationary cUTPs is located in the center of the quarter. There are two delivery zones of 1 km$^2$ each.

3.　S-o: The sUTPs are optimized for each CEP provider with a delivery zone of 0.25 to 2 km$^2$. Semi-stationary swap bodies are used for deliveries by trucks. Stationary UTPs are used for deliveries by vans.

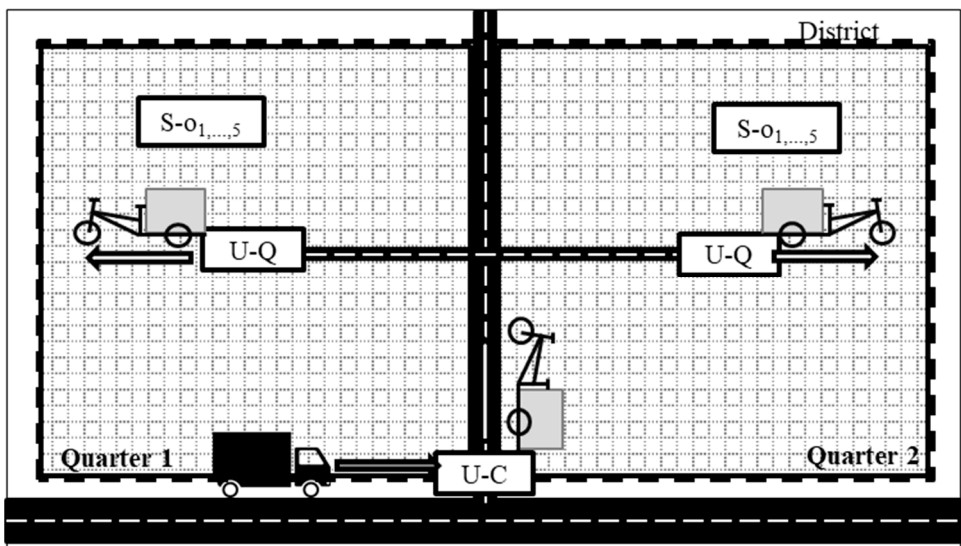

**Figure 2.** UTP-scenarios and an urban district.

Assuming that each CEP provider operates independently, and the municipality does not provide a central location, the resultant siting in S-o is suboptimal. The S-o strategy seeks the best-fitting distribution of UTPs for each logistics provider by employing an optimization method that seeks optimal UTP utilization.

*2.3. Network Vehicle Configuration*

The model replicates a straightforward delivery system. Returns and pick-ups are not factored in. Time windows are also excluded. Since the delivery van market is currently putting highly innovative electric vehicles into operation, it makes sense to analyze them too.

Parcels for the UTP network first need to be brought from the distribution center to the UTP site. The swap bodies for semi-stationary sUTPs have to be transported by heavy 12-ton trucks. Trucks or vans can transport parcels to stationary cUTPs. Swap body sUTPs must be returned to the distribution center in the evening. Round-trip deliveries follow four steps: A vehicle (van) is loaded at the distribution center; the vehicle (van) is driven into the city; deliveries are completed as a nearly optimal round trip without time windows; undeliverable parcels are returned to a drop-off point and the vehicle returns to the starting point.

In the UTP strategies, cargo bikes may return for a new round trip upon completing their first round trip as long as the maximum hours of work are not exceeded. The cargo bikes are tricycles and are assumed to have identical capacity for all the CEP services. The following network vehicle configurations (NVC) exist:

1.　Van: Deliveries are completed by conventional (diesel) vans.
2.　E-van: Deliveries are completed by electric vans.
3.　Cargo bike and trucks: Parcels are transported to the UTP either by a truck hauling swap bodies or by box trucks. Deliveries are completed by cargo bike.
4.　Cargo bike and van: A van brings the parcels to the UTPs, sUTPs are considered small stationary objects. Deliveries are completed by cargo bike.
5.　Cargo bike and e-van: An electric van delivers the parcels to the UTPs. Deliveries are completed by cargo bike.

It is assumed that the vehicles completing the inbound and outbound trips either stay at the UTP or complete deliveries/pick-ups at other locations in the city, and that they therefore enter and leave the city just once a day.

## 3. Mathematical Formulation of the Model

The mathematical model is constructed as an analytical, deterministic representation of the logistics network from the distribution center to the receivers with its different UTP strategies. It is static, calculating the impact of one average day based on fixed inputs. The logistics processes in the network are modelled as a set of related algebraic equations. Table 1 provides a full list of notations.

**Table 1.** List of notations.

| Parameter | Description |
|---|---|
| A | area of a delivery zone Z |
| D | total distance of a strategy |
| $d_{DC}$ | distance to depot |
| $D_{del}$ | total distance delivery |
| $d_{LSP}$ | delivery distance per CEP-company /LSP |
| $d_{r, UTP}$ | final tour serving multiple UTP |
| $D_S$ | total distance in- and outbound process |
| $d_s$ | in- and outbound distance |
| ET | emission type |
| $f_{fit}$ | fitting factor |
| $f_{LSP}$ | market share of a CEP-company |
| $f_P$ | percentage of parcels suited for cargo bikes (50%) |
| $f_{Stop}$ | stop factor |
| $n_a$ | delivery zone factor |
| P | parcel volume |
| p | parcels per resident |
| Q | capacity |
| $\bar{r}_A$ | average distance to receiver |
| SB | swap body |
| STR | strategy to site an urban transshipment point |
| $t_{load}$ | loading time |
| $t_{max}$ | maximum permitted working hours |

### 3.1. General Problem Formulation

Let $ET = \{GHG; NOx; PM10\}$ denote a triplet of emission types, $STR = \{U\_C; U\_Q; S\_o\}$ a triplet of UTP-strategies, and $NVC = \{Van; EVan; CB; CB\_Van; CB\_EVan\}$ a set of possible vehicle network configurations. The quantity of type $ET$ emissions and the UTP strategy $STR$ is calculated with $D_S^{(STR),\,(NVC)}$ as the total distance traveled for the inbound and outbound processes and $D_{del}^{(STR),\,(NVC)}$ is the total distance of all vehicles of each strategy and NVC for the delivery processes (Equation (1)). Both parameters are calculated as a function of the parcel volume ($P$). The emissions are obtained by multiplying distances by emission factors ($EF_{ET}$), which are also a function of the $NVC$. This is based on [42].

$$E_{ET}^{(STR),\,(NVC)} = D_S^{(STR),\,(NVC)}(P) \times EF_{ET}^{(NVC)} + D_{del}^{(STR),\,(NVC)}(P) \times EF_{ET}^{(NVC)} \tag{1}$$

### 3.2. Delivery Process Formulation

The calculation of deliveries follows a two-step procedure. The first step initializes the UTP strategy and assigns the parcel volume $P$ to the delivery zone $Z$. A zone is a part of the delivery area served by one UTP entity. The parcel volume is a function of parcels per resident ($p_{y,\,d}$) for the year ($y$)

being analyzed and the type of day ($d$), the size of the delivery zone ($A_Z^{(STR)}$), the percentage of parcels actually suited for cargo bikes ($f_p^{(LSP)}$), and each logistics provider's market share ($f_{LSP}$).

$$P_Z^{(STR),\ (LSP)} = p_{y,d} \times A_z^{(STR)} \times f_p^{(LSP)} \times f_{LSP} \tag{2}$$

The values for $p_{y,d}$ are specified in Section 4. The value of $f_p^{(LSP)}$ is set at 0.5. This represents an average value for comparable urban forms identified in talks with industry experts. The superscript index LSP, however, indicates that this factor is a function of each logistics provider's policy and parcel structure. The market share may change.

Equation (3) calculates the distance each logistics provider travels for a given strategy and NVC ($d_{LSP}^{(STR),(NVC)}$). The methodology is a continuous approximation of the trip distance following the Daganzo heuristic [43]. This results in nearly optimal routes and is applicable to problems with more than six nodes per stop. This methodology is applied in comparable studies [44–46]. Using Daganzo's heuristic is advisable since the nodes (number of stops) are discrete locations distributed uniformly in the delivery zone. The number of stops is computed by dividing the number of parcels for a zone $P_Z^{(STR),\ (LSP)}$ by the stop factor ($f_{stop}$), assuming that there can be more than one receiver per stop. Parcel volume divided by delivery vehicle capacity ($Q_{NVC}$) yields the number of trips necessary. The factor $\bar{r}_A^{(STR)}$ specifies the average distance between the route's starting point and the nodes that constitute stops. It is calculated as a fixed value beforehand. The factor $f_{fit}$ is a fitting and detour factor that can be used to adjust the distance to real-world data. Up until this step, the formula only delivers the distance driven in one delivery zone. Since the strategies allow smaller and thus more delivery zones, $n_A$ is used to add the distance over all zones per strategy. The factor $n_A$ is calculated by dividing the size of the delivery area (2 km²) by the size of the delivery zone $A_Z^{(STR)}$.

$$d_{LSP}^{(STR),\ (NVC)} = \left( 2 * \bar{r}_A^{(STR)} \times \frac{P_Z^{(STR),\ (LSP)}}{Q_{NVC}} + 0,57 \sqrt{\frac{P_Z^{(STR),\ (LSP)}}{f_{Stop}} \times A_Z^{(STR)}} \right) \times f_{fit} \times n_A \tag{3}$$

$D_{del}^{(STR),\ (NVC)}$ is obtained by totaling $d_{LSP}^{(STR),\ (NVC)}$ for all logistics providers. The factor $f_{Stop}$ related to the population scenarios makes it possible to calculate the number of receivers at stops (Table 2).

**Table 2.** Population scenario trip characteristics.

| Variable | Population Scenario | Vehicle | Value | Unit |
|---|---|---|---|---|
| Parcels per stop | Mixed-use | Van | 3.1 | p/stop |
| | | Cargo bike | 1.6 | p/stop |
| | Residential | Van | 1.7 | p/stop |
| | | Cargo bike | 1.3 | p/stop |
| Parcels per receiver | Mixed-use | Van | 1.3 | p/receiver |
| | | Cargo bike | 1.3 | p/receiver |
| | Residential | Van | 1.2 | p/receiver |
| | | Cargo bike | 1.2 | p/receiver |
| Stop duration | Mixed-use | Van | 7.2 | min./stop |
| | | Cargo bike | 3.6 | min./stop |
| | | Van | 4.2 | min./stop |
| | Residential | Cargo bike | 3 | min./stop |

Since more than one round trip is necessary to deliver all the parcels, a single vehicle may make more than one trip. The number of trips is restricted by the maximum permissible hours of work ($t_{max}$ = 7.5 h excluding breaks). $v_{NVC}$ defines the average speed. We assume that a vehicle's period

of operation equals a courier's hours of work. The time for one round trip $t_{r,Z}^{(STR),\,(NVG),(LSP)}$ can be calculated with Equation (4). The term $t_{stop}$ is the average duration of a stop in a population scenario (Table 1). The time loading and unloading for a trip by cargo bike is assumed to be 0.4 h. It is 2 h for vans, including the inbound and outbound processes.

$$t_{r,Z}^{(STR),\,(NVG),(LSP)} = \left( \frac{d_{LSP}^{(STR),\,(NVC)}}{v_{NVC}} + \frac{P_Z^{(STR),\,(LSP)}}{f_{Stop}} \times t_{Stop} \right) \times \frac{Q_{NVC}}{P_Z^{(STR),\,(LSP)}} + t_{load} \tag{4}$$

The S-o strategy is slightly different. Each logistics provider desires to use the optimal (minimum) number of swap body sUTPs. This means that a new sUTP is only opened when the capacity of the preceding configuration has been exceeded. Its opening is concomitant with the reduction of the size of the delivery zone to improve delivery effectiveness. This process is reproduced in Equation (5), which employs discrete values of delivery zone size $A = \{0, 25; 0, 5; 1; 2\}$ to seek a minimum of unutilized capacity in the swap bodies ($\Delta Q$). This keeps the results comparable with the other strategies. $Q_{SB}$ is the swap body's capacity of 400 parcels.

$$\min \Delta Q = 1 - \frac{P_Z^{(STR),\,(LSP)}}{Q_{SB}} \tag{5}$$

### 3.3. Formulation of the Inbound and Outbound Processes

The inbound and outbound processes are often neglected when assessing cargo bike systems. We include them by incorporating the total distance traveled $D_S^{(STR),\,(NVC)}$ either to supply the UTPs or for the vans to reach the delivery zone for each population density, UTP strategy, and NVC. This is the sum of the distance $d_S^{(STR),\,(NVC),(LSP)}$ traveled by each logistics provider in each strategy and with each vehicle configuration (Equation (6)). The average distance between the logistics providers' distribution centers and the delivery area $d_{DC}$ is assumed to be 10 km and denotes the distance traveled outside the district. Since most strategies site the UTPs inside the district, assessing the kilometrage associated with the supply process within the district is also useful. The delivery distance is not calculated for the baseline case. The UTPs' delivery distance within the district is represented by $d_{UTP}^{(STR)}$, which is the distance between the individuals UTPs and the entrance to the district for each strategy. Vans and box trucks serving the UTPs are assumed to be fully loaded. The last truck makes deliveries to the UTPs on a round trip, represented by $d_{r,UTP}^{(STR)}$. The number of trips necessary is determined by dividing the parcel volume per logistics provider and strategy by the capacity $Q_{NVC}$ of the vehicle in operation. Vehicle capacities are listed in Table 2.

$$d_S^{(STR),\,(NVC),(LSP)} = \left( 2 \times d_{DC} + 2 \times d_{UTP}^{(STR)} \right) \times \frac{P_Z^{(STR),\,(LSP)} \times n_A}{Q_{NVC}} + d_{r,UTP}^{(STR)} \tag{6}$$

### 3.4. Implementation and Validation

The modeling tool of choice is Microsoft Excel. It enables a direct and unambiguous representation of the formal model for each UTP-strategy. The strategies are modelled in a parallel manner so that they can be computed all at one. To control our input parameters reflecting population density scenario and network vehicle configuration a scenario manager is set up. Thus, population density scenarios and network vehicle configuration can be adjusted independently. Each population density scenario therefore forms its own experiment with a set of all possible NVCs applied to the implemented UTP strategies. Thus, we create the notation NVC-UTP-strategy (e.g., Truck–U-C). The tool then runs the experiment through a VBA-script modifying the parcel volumes in steps of 5000 parcels/d within the defined range. Results are automatically plotted in separate worksheets.

During the research project "Lastenraddepot" [15], delivery tours modelled with our tool were checked against performance indicators of real-world applications and proved to be valid. The real-world delivery characteristics and vehicle capacities presented in [6,47] are used for validation.

## 4. Data Sets

Although the CEP industry has established actors, data on freight demand and delivery characteristics (e.g., route performance, costs) are scarce. The model therefore focuses on external impacts without analyzing expenditures and revenues. The exact number of parcels per quarter is calculated using population density (in steps of 5000 p/km$^2$) in order to develop reliable demand scenarios. The total parcel volume of 3.16 million per year (based on Germany in 2016) is taken from [48]. Sixty percent are business-to-customer deliveries and 40% are business-to-business deliveries. This ratio is similar to that of businesses and residents in mixed-use quarters [47]. The number of parcels per mixed-use quarter can thus be calculated using the residents per km$^2$, thus reflecting density's effect. The method must be modified for residential quarters by applying the factor of 0.7. An annual increase of 5.4% [48] results in 0.17 parcels per resident in the year 2025.

The five CEP services are differentiated by market share [6]. One company has a market share of nearly fifty percent, while the other companies divide the remaining market share among themselves roughly equally. This distribution can be modified for local circumstances as necessary.

The data used to calculate the GHG, NOx, and PM10 emissions of a Euro 6 7.5–12 ton truck and an N-III Euro 6d light duty vehicle are taken from Handbook Emission Factors for Road Transport (HBEFA 4). These were chosen because they are likely to reflect a fleet mix in 2025.

To represent the different population scenarios, we implemented different trip characteristics (Table 2). The values for vans are real-world data taken from [47]. The values for cargo have been adjusted accordingly.

The capacity values for each vehicle of the NVCs (QNVC) estimated are based on industry experience [6,47] (Table 3). The difference in capacity between trucks with swap bodies and box trucks arises from the need for presorting inside swap bodies at the distribution center. Box trucks can be loaded more densely since space does not have to be left for presorting.

**Table 3.** Vehicle capacity.

| Vehicle | QNVC | Unit |
|---|---|---|
| Van (diesel and electric) | 160 | Parcels/vehicle |
| Cargo bike | 40 | Parcels/vehicle |
| Truck (swap body) | 400 | Parcels/vehicle |
| Truck (box) | 1000 | Parcels/vehicle |

## 5. Experiments and Results

The objective of this study was to assess the impacts of UTP siting and NVC selection on urban traffic and emissions. This is pursued in-depth for the experiment of the mixed-use business scenario. The impact of the population scenario and parcel density per resident are discussed in the following. The impact on traffic is presented in Table 4. As prior studies established, the use of cargo bikes instead of vans to make deliveries actually increases CEP providers' kilometrage within the district because the bikes travel back and forth to the UTPs (Figure 3). The S-o strategy employing distributed UTPs has the lowest cargo bike delivery kilometrage since UTPs are at the closest location to the receivers. The analysis of inbound and outbound processes reveals that the U-Q and S-o strategies require trips with trucks within the district. S-o shows a strong increase at this point with raising parcel numbers. Trucks reduce kilometrage overall because of the impact of the trucks' capacity on consolidation. Using vans to make deliveries to UTPs translates into an overall increase of kilometrage. The reasons for that are missing consolidation for in- and outbound processes and the usage of cargo bikes for deliveries.

**Table 4.** Impacts on traffic and kilometrage, "district" is defining in- and outbound kilometrage within the district.

| Residents/km² | | | Conv | Truck-UTP | | | Van-UTP | | |
|---|---|---|---|---|---|---|---|---|---|
| | P/d | | Van | U-C | U-Q | S-o | U-C | U-Q | S-o |
| 10,000 | 1800 | delivery | 119.4 | 209.6 | 176.2 | 179.2 | 209.6 | 176.2 | 179.2 |
| | | district | 0.0 | 0.0 | 15.0 | 15.0 | 0.0 | 33.0 | 32.0 |
| | | total | 399.4 | 309.6 | 291.2 | 334.2 | 489.6 | 489.2 | 491.2 |
| 15,000 | 2700 | delivery | 149.0 | 273.9 | 222.2 | 228.9 | 273.9 | 222.2 | 228.9 |
| | | district | 0.0 | 0.0 | 17.0 | 16.0 | 0.0 | 43.0 | 41.5 |
| | | total | 529.0 | 393.9 | 359.2 | 404.9 | 653.9 | 645.2 | 650.4 |
| 20,000 | 3600 | delivery | 178.3 | 331.8 | 266.4 | 254.1 | 331.8 | 266.4 | 254.1 |
| | | district | 0.0 | 0.0 | 17.0 | 28.0 | 0.0 | 57.0 | 68.0 |
| | | total | 698.3 | 451.8 | 403.4 | 562.1 | 851.8 | 843.4 | 842.1 |
| 25,000 | 4500 | delivery | 199.2 | 391.1 | 304.5 | 283.1 | 391.1 | 304.5 | 283.1 |
| | | district | 0.0 | 0.0 | 19.0 | 32.0 | 0.0 | 63.0 | 85.5 |
| | | total | 779.2 | 531.1 | 463.5 | 635.1 | 971.1 | 947.5 | 948.6 |
| 30,000 | 5400 | delivery | 224.7 | 445.3 | 343.5 | 316.0 | 445.3 | 343.5 | 316.0 |
| | | district | 0.0 | 0.0 | 19.0 | 32.0 | 0.0 | 77.0 | 101.0 |
| | | total | 944.7 | 585.3 | 502.5 | 668.0 | 1165.3 | 1140.5 | 1137.0 |
| 35,000 | 6300 | delivery | 245.9 | 499.3 | 379.3 | 342.3 | 499.3 | 379.3 | 342.3 |
| | | district | 0.0 | 0.0 | 21.0 | 48.0 | 0.0 | 87.0 | 122.5 |
| | | total | 1065.9 | 659.3 | 560.3 | 870.3 | 1319.3 | 1286.3 | 1284.8 |

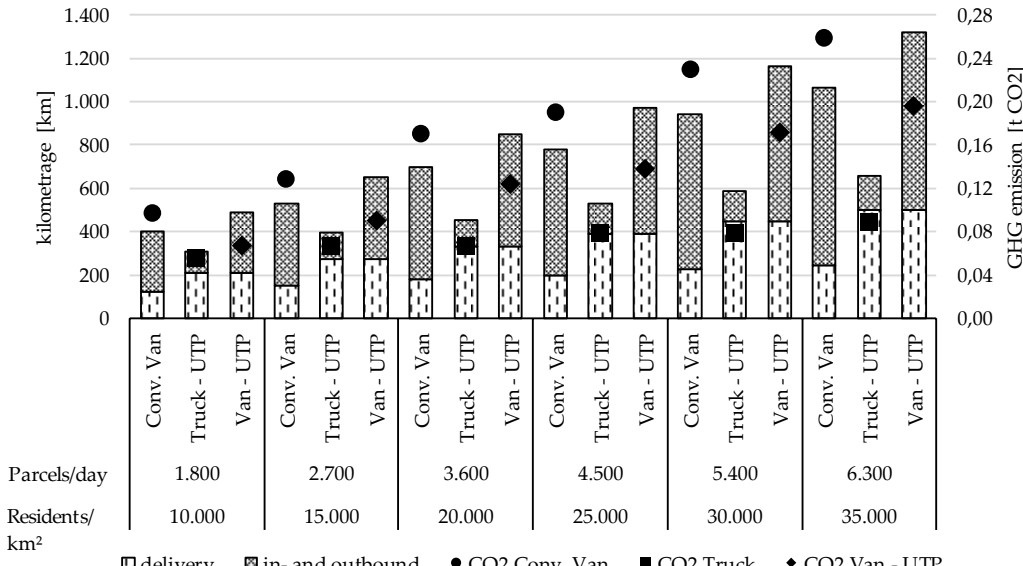

**Figure 3.** Kilometrage and greenhouse gas (GHG) emissions; comparison of conventional delivery and UTP strategy U-C with trucks and vans.

Since GHG's are global emissions, they are calculated for the complete system. The results (Table 5) clearly reveal that cargo bikes reduce emissions by as much as 66% over deliveries by conventional vans. There is an important difference between the UTP strategies and the NVC. The cUTP and thus Q-C strategies' greater potential to cut costs correlates with the lowest kilometrage. The S-o strategy does not necessarily reduce GHG emissions when trucks are used. The use of trucks and high capacity swap bodies establishes constraints under which such a system may not be beneficial.

**Table 5.** Carbon dioxide ($CO_2$) emissions and potential reductions in ton of $CO_2$ per day for a mixed-use district scenario.

| Residents/km² | P/d/km² | Baseline | | Truck-UTP | | | Van-UTP | | | e-Van-UTP | | |
|---|---|---|---|---|---|---|---|---|---|---|---|---|
| | | Van | e-Van | U-C | U-Q | S-o | U-C | U-Q | S-o | U-C | U-Q | S-o |
| 10,000 | 1800 | 0.098 | 0.062 | 0.056 | 0.065 | 0.088 | 0.067 | 0.075 | 0.075 | 0.045 | 0.050 | 0.050 |
| | | | | −43%/−10% | −34%/5% | −11%/42% | −32%/8% | −24%/21% | −24%/21% | −55%/−29% | −50%/−20% | −50%/−21% |
| 15,000 | 2700 | 0.129 | 0.082 | 0.068 | 0.077 | 0.100 | 0.091 | 0.101 | 0.101 | 0.060 | 0.067 | 0.067 |
| | | | | −48%/−19% | −41%/−7% | −23%/22% | −30%/11% | −22%/23% | −23%/23% | −54%/−27% | −49%/−19% | −49%/−19% |
| 20,000 | 3600 | 0.171 | 0.109 | 0.068 | 0.077 | 0.175 | 0.125 | 0.138 | 0.141 | 0.083 | 0.092 | 0.093 |
| | | | | −61%/−38% | −55%/−30% | 3%/61% | −27%/15% | −19%/27% | −18%/30% | −52%/−25% | −47%/−16% | −46%/−15% |
| 25,000 | 4500 | 0.190 | 0.122 | 0.079 | 0.090 | 0.200 | 0.139 | 0.154 | 0.159 | 0.092 | 0.102 | 0.106 |
| | | | | −59%/−36% | −53%/−27% | 5%/65% | −28%/15% | −20%/27% | −17%/31% | −52%/−25% | −47%/−17% | −45%/−14% |
| 30,000 | 5400 | 0.230 | 0.148 | 0.079 | 0.090 | 0.200 | 0.172 | 0.191 | 0.197 | 0.114 | 0.127 | 0.130 |
| | | | | −66%/−47% | −62%/−40% | −14%/36% | −26%/17% | −18%/30% | −15%/34% | −51%/−23% | −46%/−15% | −44%/−12% |
| 35,000 | 6300 | 0.260 | 0.167 | 0.090 | 0.102 | 0.299 | 0.196 | 0.217 | 0.226 | 0.130 | 0.144 | 0.150 |
| | | | | −66%/−46% | −61%/−39% | 16%/80% | −25%/18% | −17%/31% | −14%/36% | −50%/−22% | −45%/−14% | −43%/−11% |

The interplay between kilometrage and GHG emissions is displayed in Figure 3 using U-C strategy as example. The van-UTP case compared to the conventional delivery clearly demonstrate a contradicting tendency. Decreasing GHG emissions means increasing kilometers travelled. It is important to mention that the van distance for in- and outbound processes is equal. The effect in both measurements is purely caused by substitution of vans through cargo bikes for deliveries within the district. Looking at the truck-UTP case, both kilometrage and GHG emission decrease. The difference between van and truck forms the emission factor applied. Higher GHG-emissions per kilometer with trucks obviously do not outweigh the consolidation effect of higher capacity and consequently less tours necessary.

Electric vans change the picture slightly (Table 5); cUTPs served by trucks still have greater potential to reduce GHG emissions compared to the conventional system with e-vans. The difference appears when e-vans are performing in- and outbound processes of the UTPs. A threshold exists between smaller and larger amounts of parcels. E-vans reduce GHG emissions relatively more in the first case, trucks work well for larger amounts. This reveals that economies of scale are a driving force. The potential reduction of GHG emissions generally increases as the parcel volume increases when trucks are used.

Cargo bikes are considered a means to reduce PM10 in affected districts. Our results, however, indicate that this target is largely dependent on the vehicle used to service the UTP and on the UTP's actual location (see Table 6). Siting a cUTP at a district's periphery logically reduces PM10 emissions 100% since no vans or trucks enter the district. Trucks especially need to be studied thoroughly since their PM10 emissions factor is approximately ten times higher than the one for vans. In contrast to GHG emissions the high emission factor neutralizes the trucks' consolidation effect. cUTPs in districts or even S-o strategies could elevate PM10 emissions by 53% for U-Q or even by as much as 137% for S-o. Unlike trucks, vans have great potential to reduce PM10 emissions. Vans used for inbound and outbound processes can reduce emissions by as much as 63% over the electric van baseline.

Analyzing $NO_x$ emissions produces similar results when e-vans are not considered (see Table 7). Since electric vehicles emit no NOx, servicing a UTP with electric vans could reduce emissions 100% in any UTP strategy. Apart from this, NVC in which vans perform in- and outbound processes strongly decrease NOx in all cases of U-Q and S-o. Trucks are different. The S-o strategy results in no reduction. For U-Q strategies again a threshold applies. Below 20,000 residents/km$^2$ conventional vans lead to less NOx, above that value trucks improve air quality.

The impact of the business scenario is determined with the aid of $CO_2$ emissions (Table 8). Implementing cargo bike deliveries in residential areas has a greater impact than the mixed-use scenario since the lower density of parcels per stop increases the distance traveled per delivery route within the district. Whereas this results in higher emissions for vans, cargo bikes can utilize their potential as low-emission vehicles.

**Table 6.** Particulate matter (PM10) emissions and potential reductions in grams of PM10 per day in a mixed-use district.

| Residents/km² | P/d/km² | Baseline Van | Baseline e-Van | Truck-UTP U-C | Truck-UTP U-Q | Truck-UTP S-o | Van-UTP U-C | Van-UTP U-Q | Van-UTP S-o | e-Van-UTP U-C | e-Van-UTP U-Q | e-Van-UTP S-o |
|---|---|---|---|---|---|---|---|---|---|---|---|---|
| 10,000 | 1800 | 4.94 | 4.67 | 0.00 −100%/−100% | 7.52 53%/62% | 7.52 53%/62% | 0.00 −100%/−100% | 1.37 −73%/−71% | 1.32 −74%/−72% | 0.00 −100%/−100% | 1.16 −77%/−76% | 1.12 −78%/−77% |
| 15,000 | 2700 | 6.17 | 5.82 | 0.00 −100%/−100% | 8.52 39%/47% | 8.02 30%/38% | 0.00 −100%/−100% | 1.78 −72%/−70% | 1.72 −73%/−71% | 0.00 −100%/−100% | 1.51 −76%/−75% | 1.45 −77%/−76% |
| 20,000 | 3600 | 7.38 | 6.96 | 0.00 −100%/−100% | 8.52 16%/23% | 14.03 91%/102% | 0.00 −100%/−100% | 2.36 −69%/−67% | 2.82 −62%/−60% | 0.00 −100%/−100% | 2.00 −73%/−72% | 2.38 −68%/−66% |
| 25,000 | 4500 | 8.25 | 7.76 | 0.00 −100%/−100% | 9.52 16%/23% | 16.03 95%/107% | 0.00 −100%/−100% | 2.61 −69%/−67% | 3.54 −58%/−55% | 0.00 −100%/−100% | 2.21 −74%/−72% | 2.99 −64%/−62% |
| 30,000 | 5400 | 9.30 | 8.75 | 0.00 −100%/−100% | 9.52 3%/9% | 16.03 73%/84% | 0.00 −100%/−100% | 3.19 −66%/−64% | 4.18 −56%/−53% | 0.00 −100%/−100% | 2.70 −72%/−70% | 3.54 −62%/−60% |
| 35,000 | 6300 | 10.18 | 9.57 | 0.00 −100%/−100% | 10.52 4%/10% | 24.05 137%/152% | 0.00 −100%/−100% | 3.60 −65%/−63% | 5.07 −51%/−48% | 0.00 −100%/−100% | 3.05 −71%/−69% | 4.29 −58%/−56% |

**Table 7.** Nitrogen oxide (NOx) emissions and potential reductions in grams of NOx $g_{NOx}$ per day, mixed-use district.

| Residents/km² | P/d | Conventional | | Truck-UTP | | | Van-UTP | | |
|---|---|---|---|---|---|---|---|---|---|
| | | Van | e-Van | U-E | U-Q | S-o | U-E | U-Q | S-o |
| 10,000 | 1800 | 7.400 | 0.000 | 0.000 | 9.000 | 9.000 | 0.000 | 2.046 | 1.984 |
| | | | | −100% | 22% | 22% | −100% | −73% | −74% |
| 15,000 | 2700 | 9.24 | 0.00 | 0.00 | 10.20 | 9.60 | 0.00 | 2.67 | 2.57 |
| | | | | −100% | 11% | 4% | −100% | −72% | −73% |
| 20,000 | 3600 | 11.05 | 0.00 | 0.00 | 10.20 | 16.80 | 0.00 | 3.53 | 4.22 |
| | | | | −100% | −8% | 53% | −100% | −69% | −62% |
| 25,000 | 4500 | 12.35 | 0.00 | 0.00 | 11.40 | 19.20 | 0.00 | 3.91 | 5.30 |
| | | | | −100% | −8% | 56% | −100% | −69% | −58% |
| 30,000 | 5400 | 13.93 | 0.00 | 0.00 | 11.40 | 19.20 | 0.00 | 4.77 | 6.26 |
| | | | | −100% | −19% | 38% | −100% | −66% | −56% |
| 35,000 | 6300 | 15.24 | 0.00 | 0.00 | 12.60 | 28.80 | 0.00 | 5.39 | 7.60 |
| | | | | −100% | −18% | 89% | −100% | −65% | −51% |

**Table 8.** Residential and mixed-use population scenarios for GHG emissions in $t_{CO2}$ per day.

| Residents/km² | | Baseline | | | | Truck | | Van | |
|---|---|---|---|---|---|---|---|---|---|
| | | Mixed | | Resident. | | Mixed | Resident. | Mixed | Resident. |
| | P/d | Van | e-Van | Van | e-van | U-Q | U-Q | U-Q | U-Q |
| 10,000 | 1800 | 0.098 | 0.062 | 0.088 | 0.055 | 0.065 | 0.065 | 0.075 | 0.059 |
| 15,000 | 2700 | 0.129 | 0.082 | 0.120 | 0.076 | 0.077 | 0.077 | 0.101 | 0.085 |
| 20,000 | 3600 | 0.171 | 0.109 | 0.147 | 0.093 | 0.077 | 0.077 | 0.138 | 0.107 |
| 25,000 | 4500 | 0.190 | 0.122 | 0.183 | 0.116 | 0.090 | 0.077 | 0.154 | 0.138 |
| 30,000 | 5400 | 0.230 | 0.148 | 0.203 | 0.129 | 0.090 | 0.090 | 0.191 | 0.154 |
| 35,000 | 6300 | 0.260 | 0.167 | 0.239 | 0.152 | 0.102 | 0.090 | 0.217 | 0.186 |

*5.1. Sensitivity Analysis*

When constructing the model, we assumed that the variables of vehicle capacity and the distance between UTPs and the distribution center were fixed values. We conducted a sensitivity analysis to test their impact. We devised a minimum and a maximum scenario in which the parameters are changed by 50% to reduce or to increase emissions. Thus, worsening the set-up is to decrease the capacity of the vehicles by 50% and to increase the distance between depot and UTPs by 50%. Improving the situation is vice versa. Adjusting capacity represents either smaller vehicles used, or larger parcels delivered.

Our findings for GHG emissions (Figure 4) indicate that conventional delivery systems using standard vans are highly sensitive to parameter changes, especially reducing capacities and increasing outbound distance. Interestingly the S-o strategy instead of conventional delivery systems with e-vans shows a similar pattern, resulting from the high kilometrage for in- and outbound processes. Apparently, U-C and U-Q strategies equipped with trucks show a comparable low sensitivity. Thus, a system with larger parcels in average will not impact the carbon footprint as strongly as corresponding van cases. Concerning GHG-emissions, we can state that both cUTP strategies deployed with trucks decrease them effectively over a broad range of parameter settings.

The sensitivity of PM10 and NOx emissions differs from the findings above. U-C strategies and e-vans are fixed at zero emissions within the district (e-van just for NOx). Worsening the situation results in minor impacts in all other cases, despite S-o strategies. Due to high numbers of truck driving levels within the district a strong impact is seen when decreasing load capacity/increasing parcel volume in truck NVC.

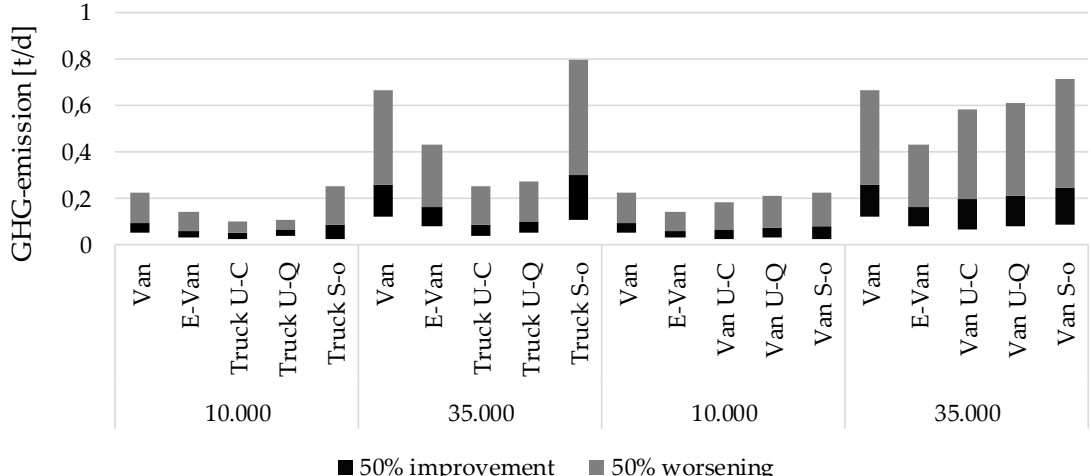

**Figure 4.** Sensitivity analysis of GHG emissions, only minimum and maximum population densities displayed.

*5.2. Summery and Critical Reflection*

Our model clearly demonstrates that cargo bike delivery networks can improve urban livability significantly by improving air quality, reducing GHG emissions, and van traffic in urban districts. Apart from this general finding, some aspects need close consideration to avoid negative effects. The critical factor is the selection of a network vehicle configuration. It is a decision between highly reducing GHG emissions and reducing other emissions (PM10, NOx). On the one hand using trucks to service UTPs strongly reduces GHG emissions. On the other hand, vans produce stronger impact on air quality and just slightly benefit mitigating GHG emissions.

When designing a cargo bike network, it is important to consider thresholds of truck configurations. Air pollution impacts within the district are dependent on the parcel volumes. For small parcel amounts, trucks are likely to increase air pollution until a certain threshold compared to the conventional set up. In case air pollution is the major concern, planners are strongly advised to investigate the usage of electric vans, although they should not expect a strong effect on GHG-emissions. The strategies of siting distributed UTPs (S-o) is not a feasible solution when trucks and swap bodies are used. Most likely, their effects are minor or even worsen the situation. If S-o strategies are followed, vans—either electric or not—perform better than trucks and make a difference towards a more sustainable district and city.

Asking for unambiguous advice, we emphasize siting a cUTP at a district's periphery and having trucks performing the in- and outbound processes. This strategy and NVC enables consolidation effects, cuts GHG emissions and kilometrage in in- and outbound processes. Furthermore, by not driving into the district itself it avoids PM10 and NOx emissions there. Vans are viable too but will not result in the same level of GHG reductions. Our recommendation is, create candidate sites at the periphery of a district.

Our findings based on the proposed model have some limits based on the methodology and model itself. Firstly, we are solving the vehicle routing problem through the Daganzo heuristic, which gives near-optimal results but does not fully represent the topology of street networks in a European city. Secondly, the calculation of necessary UTPs by means of fixed capacities in the S-o strategies in some cases leads to very low utilization. Thus, our results are a conservative estimate. Thirdly, we assume a rectangular district with evenly distributed demand, while real-world districts are dominantly polygons resulting in less optimal routes and clusters of demand. Lastly, our insights are based on an average day of a year. Thus, we have no insights about the system behavior under fluctuating demand. The sensitivity analyses give an indication that our recommendations are still valid but do not represent situations of under- or overcapacity.

### 6. Discussion and Conclusions

We constructed an algebraic, deterministic model in our study to analyze the impact of high-volume cargo bike systems on traffic, and on GHG and other emissions. Our model clearly demonstrated that cargo bikes can improve urban livability significantly by improving air quality, GHG emissions, and van traffic in urban districts. A few other factors apart from these general and often touted impacts are significant.

The results show that implementing cargo bikes is not an automatically improving criteria of livability. We cannot confirm the results of Melo and Baptista [17] that a more than 10% increase of cargo bikes will worsen traffic, since our substitution rate of parcels is at 50%. However, we must admit that certain strategies and thresholds exist which are not benefiting. Furthermore, the methodology applied in our article assumes free flow of vehicles and consequently does not consider interaction between entities of the flows on those impacts. The latter aspect is of high importance, since the proposed cUTPs are not just consolidation flows of goods but also of vehicles, a fact often neglected in planning [49]. Since we compute PM10 and NOx on district level, we can assess benefits on this scale, but on the micro scale air pollution can worsen around a cUTP when several vans/trucks are attracted to this location. This facet needs more investigation and is also dependent on the spatial organization of depots, districts, and transport systems of each district and city. One approach can be a microscopic traffic simulation of UTPs and their flows. This would add to the valuable work of Melo and Baptista [17].

As pointed out in the introduction, cargo bike delivery networks are dominantly seen as a means to improve livability. One facet of this is reduction of traffic. Already, Browne et al. [7] made the experience of increased kilometers traveled, backing up our results. However, Arvidsson et al. [19] propose another measurement, the space occupied, and show improvements for a similar case. Although this concept adds to a public discussion about fair allocation of street space, we found it difficult to model. Space occupancy of vehicles is a product of static vehicle size and dynamic vehicle speed. The latter would need a simulation or stochastic representation of speed levels, waiting times, and matching to used links (e.g., cycling path, road). One question here would be: Is it as just to use certain space on roads for cargo bikes as on cycling paths or sidewalks? A corresponding model (e.g., in MATSIM) for simulating an entire district is worth future work.

GHG mitigating results of field trials [7,21] show reductions of approximately 50%. Our results for stationary cUTPs are similar. In contrast to those trials with a given parcel volume, we indicate the actual reductions are dependent on parcel volume and can be lower for low amounts and stronger with higher. Compared to the trial of Henrich et al. [21] facilitating semi-stationary swap bodies, our results show fewer relative reductions. This is mainly due to our calculation methodology, which does not result in fully utilized UTPs in S-o strategy. A solution can be the implementation of a variable share of parcels assigned to the cargo bike delivery network, allowing full utilization. However, this would conflict with the comparability between UTP strategies and a coherent relation between population density and parcel volumes. Within the model, it is possible to adjust cargo bike share for each CEP-LSP separately, applications on a distinct district would allow the assessment with better utilization.

The potential of cargo bikes to cut costs is not considered here, primarily because of the dearth of reliable cost data for UTPs. The best strategy for siting UTPs, i.e., U-C at a district's periphery, generates the greatest kilometrage of inner district deliveries for CEP providers. Urban planners ought to respond to this drawback by providing affordable land for cUTPs at suitable sites. The distance to customers employed in the model should not be increased. Nevertheless, future work should integrate costs from the LSP perspective to take cost-benefit factors of those into account. As Arnold et al. [20] demonstrate for a different delivery network setup, applying a total cost approach internalizing external costs of emissions can furthermore provide a better understanding of the feasibility and sustainability of the proposed strategies. Options of self-pick up should be integrated in future work.

We aimed to develop an easy-to-use model to assist urban planning. What we cannot present is a practical application and validation of its usability. This would only make sense in combination

with a field trial and sound before–after evaluation based on real-world data. We are aiming to give such an evaluation in the near future. Until the necessary field trial comes to pass, we remain to give the straightforward advice for urban planners to create space for UTPs at the district's periphery if the major objective is reduced GHG emissions and air pollution. Our contribution, compared to mentioned articles on designing urban logistics networks, gives a sound recommendation about where to create candidate locations. Since municipalities strive for the common good instead of profit [50] and are equipped with regulatory power, the missing costs consideration is of less importance from a public perspective.

Concluding, we can state that the objective to assess certain strategies of siting UTPs is met. We also provide the recommendation for strategic urban planning to reserve and create locations at the district's periphery, which is within the municipality's objective. However, the effect on air pollution at the very location needs additional consideration and the practical application of the tool remains to be proven.

**Author Contributions:** Conceptualization, T.A. and F.M.; methodology, T.A. and S.L.; model and formal analysis, T.A. and S.L.; validation, T.A., F.M., and S.L.; resources, T.A.; writing—original draft preparation, T.A. and S.L.; writing—review and editing, F.M., S.L., and T.A.; supervision, M.S.; project administration, M.S. and T.A.; funding acquisition, T.A. All authors have read and agreed to the published version of the manuscript.

**Funding:** This work is part of the project "Lastenraddepot", which is funded by the German Federal Ministry of Transport and Digital Infrastructure within the National Cycling Plan 2020 (NRVP).

**Acknowledgments:** The authors want to thank the project steering committee for their valuable insights and fruitful discussions.

**Conflicts of Interest:** The funders had no role in the design of the study; in the collection, analyses, or interpretation of data; in the writing of the manuscript, or in the decision to publish the results other than checking correct grammar and spelling.

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
