# Peer review of "Impact Assessment Model for the Implementation of Cargo Bike Transshipment Points in Urban Districts"

_sustainability, doi:10.3390/su12104082_

Round 1

Reviewer 1 Report

The author tried to present a good manuscript but technically it needs (extensive) major revision. The main problem is with the organization/structure of the paper. The contributions are also missing, describe the objectives/tasks in the introduction. The literature review section is also a weak section. Methodology (solution domain is missing or provided less description) and result section are also considered as weak sections. Authors should improve English writing as well.

Abstract:

Authors should revise the abstract by describing the problem context & need at the beginning and then mention the contributions/objectives of the research. Authors should break the first sentence into multiple sentences. the outcomes & recommendations can be described at the end.

Introduction:

Although the research background and the gap/need is described in this section, but the contribution(s) and objectives of the conducted research are missing. Authors should describe the contributions and also the objectives of this research. 

Literature review:

A lot of work has been published in this field - authors should extend this section by providing/describing the latest research work - add more references.

Methodology:

the explicit description of the model is required at the beginning of this section. describe the basic concept/terms used for the work.

I suggest authors should improve the structure of this section by describing the problem domain first and then provide the solution domain. 

tables can be used to describe the parameters, used in differential equations.

The solution domain is missing or healthy (technical) description is required 

Results:

Authors should move the data set and also the description of the area (city - in this case) at the beginning of the results section. after providing the information about the dataset, describe the experiments and also the results.

more description is required for the results - if possible describe (some) results graphically. 

A paragraph is also required which describes the critical reflection of the research/paper.

Discussion & Conclusion:

more description is required for discussion. provide the real conclusion of the paper in the revised version.

Finally (important) - improve English writing of the paper and also fix the grammatically mistakes.

Author Response

Dear Reviewer,

thank you very much for your comments. Please see the attachment for replies.

Reviewer 2 Report

Remarks in attachment

Author Response

(The authors gave the same response as above.)

Round 2

Reviewer 1 Report

In this paper, the authors presented a quantitative model to analyze the impact of high-volume cargo bike systems on traffic and on GHG and other emissions. Authors claimed, their model clearly demonstrated that the cargo bikes can improve urban livability significantly by improving air quality, GHG emissions and van traffic in urban districts. 

Although the authors tried to deal with my comments efficiently and presented a quality article however I have the following minor comments and suggestions to improve further - a minor revision is required. I did not spend much time to review this time - just quickly go through.

 - authors should review the titles of headings and sub-headings - i.e. if conducted some experiments then the title of "results" should be "experiments and results"

- Move the table located at the end in Appendix 1 (Notation) to the main body of the paper and describe only important symbols and their meanings

- The explicit description is still required at the beginning of the methodology

- Clean the text again - remove the duplicate information, improve the English, maintain the flow of the paper

Good Luck

Reviewer 2 Report

I appreciate the correction made by the authors. So - accepted in the present form.

Author Response

Dear Reviewer 2,

Thank you very much for your helpful comments and the acceptance. We corrected some spelling typos. In order to meet other comments, we added a table and had to modify the table - text relation a bit to maintain the page flow.

King Regards

Tom Assmann